# Enhancing Spiritual Well-Being, Physical Activity, and Happiness in Hospitalized Older Adult Patients with Swallowing Difficulties: A Comparative Study of Thickeners and Swallowing Exercises

**DOI:** 10.3390/healthcare11182595

**Published:** 2023-09-21

**Authors:** Yu-Yin Kao, Yun-Ru Lai, Chiung-Yu Huang, Meng-Yun Tsai, Ming-Chun Kuo, Hsin-Wei Chen, Suey-Haur Lee, Chen-Hsiang Lee

**Affiliations:** 1Department of Nursing, Kaohsiung Chang Gung Memorial Hospital, Kaohsiung 83301, Taiwan; yuyin0020@cgmh.org.tw; 2Departments of Neurology, Hyperbaric Oxygen Therapy Center, Kaohsiung Chang Gung Memorial Hospital, Kaohsiung 83301, Taiwan; yunrulai@cgmh.org.tw; 3Department of Nursing, I-Shou University, Kaohsiung 82445, Taiwan; chyh@isu.edu.tw; 4Departments of Chest, Kaohsiung Chang Gung Memorial Hospital, Kaohsiung 83301, Taiwan; ck940176@gmail.com (M.-Y.T.); alevel23@yahoo.com (S.-H.L.); 5Departments of Oncology, Kaohsiung Chang Gung Memorial Hospital, Kaohsiung 83301, Taiwan; mp9382@cgmh.org.tw; 6Departments of Nephrology, Kaohsiung Chang Gung Memorial Hospital, Kaohsiung 83301, Taiwan; a955956002@cgmh.org.tw; 7Department of Internal Medicine, Division of Infectious Diseases, Kaohsiung Chang Gung Memorial Hospital, College of Medicine, Chang Gung University, Kaohsiung 83301, Taiwan; 8Department of Internal Medicine, Division of Infectious Diseases, Chiayi Chang Gung Memorial Hospital, Chiayi 61363, Taiwan

**Keywords:** dysphagia, thickener, swallowing exercises, older adult patients, comparative analysis, patient-reported outcomes

## Abstract

Swallowing difficulties often occur in older adult patients during acute hospitalization, leading to reduced nutritional intake, increased frailty, and various psychosocial challenges. This randomized controlled study aimed to assess the effects of two interventions, thickeners and swallowing exercises, on the spiritual well-being, physical activity, and happiness of older adult patients with swallowing difficulties during acute hospitalization from October 2019 to August 2020. Sample size calculation was performed using a conservative estimate approach, resulting in an estimate-required sample size of 42 participants. The sampling method was a random cluster sampling approach, with three ward rooms assigned to the thickeners group, swallowing exercises group and control group, respectively. Seventy-two participants were assigned to the intervention groups (thickeners or swallowing exercises) or the control group using a 1:1:1 stratified random assignment. Data were collected before and after the intervention, and matched samples were analyzed using t-tests, ANOVA, and generalized estimating equations for statistical analysis. Both intervention groups showed significant improvements in spiritual well-being (*p* < 0.001), physical activity (*p* < 0.001), and happiness (*p* < 0.001) compared to the control group. However, there were no significant differences between the intervention groups. Our findings suggest that interventions involving thickeners and swallowing exercises have positive effects on the spiritual well-being, physical activity, and happiness of older adult patients with swallowing difficulties during acute hospitalization and emphasize the importance of implementing these interventions to enhance the overall well-being and quality of life of this vulnerable patient population.

## 1. Introductions

Dysphagia poses a significant challenge in older adult patients during acute hospitalization and can lead to various complications, including recurrent coughing and aspiration pneumonia [1,2]. A considerable number of older adult patients admitted for acute illnesses experience swallowing difficulties, which significantly increase the risks of aspiration pneumonia and mortality [3]. Moreover, dysphagia in hospitalized older adult patients can result in malnutrition and dehydration, leading to increased hospital expenses [4]. However, the specific needs of this patient population are often overlooked in general wards, as interventional studies have primarily focused on neurological and rehabilitation wards.

Swallowing exercises play a crucial role in enhancing the safety and effectiveness of the swallowing process by facilitating the passage of food through the oral-pharyngeal region [5]. These exercises encompass various techniques, including chin tuck, oral and facial muscle exercises, effortful swallowing, tongue-holding maneuvers, and supine head-raising [6,7,8]. Extensive research suggests that implementing swallowing exercises in patients with dysphagia can lead to increased calorie intake, reduced weight loss, and improved survival rate [9,10]. In addition to swallowing exercises, thickeners increase liquid viscosity and have been shown to significantly contribute to improved calorie intake, nutritional status, and the prevention of aspiration pneumonia [11,12]. Moreover, the use of thickeners has been associated with noteworthy enhancements in swallowing-related quality of life [13].

Interventional strategies for dysphagia are frequently employed in patients with Parkinson’s disease, stroke, multiple sclerosis, and oral cancer [8,14]. However, most previous research studies have focused on evaluating the effectiveness of these strategies specifically in stroke patients, with relatively few focusing on hospitalized older adult patients with swallowing difficulties. Consequently, the effectiveness of implementing these interventions early in hospitalized older adult patients with dysphagia in terms of reducing adverse outcomes remains uncertain. To the best of our knowledge, there were no controlled studies exploring the effect of swallowing interventions on spiritual well-being, physical activity, and happiness of older adult patients with swallowing difficulties during acute hospitalization.

Research has indicated that older adult patients during acute hospitalization are at an increased risk of malnutrition by 155% when they experience swallowing difficulties, while the risk of diminished activities of daily living rises by 60% [15]. It is evident that swallowing difficulties have a substantial impact on nutritional intake and daily functioning in these population. Following a decline in activities of daily living, older adults often struggle to maintain social engagement, significantly affecting their spiritual and psychological well-being. Figueira et al. have demonstrated numerous benefits of regular physical activity interventions for older adults, not only enhancing muscle mass but also improving spiritual well-being, potentially promoting healthier aging [16]. Moreover, increased physical activity has been found to enhance social engagement and lead to more pronounced improvements in spiritual well-being [17]. Several studies have also suggested that enhanced physical activity in the elderly indirectly contributes to an improved sense of happiness [18,19]. However, it remains uncertain whether improvements in swallowing function following the onset of swallowing difficulties in older adult patients during acute hospitalization would result in a subsequent enhancement of their activity levels, spiritual well-being, and happiness. To address this knowledge gap, the aim of this study was to investigate the spiritual well-being, physical activity, and happiness impact of these interventions in older adult patients with dysphagia beyond the stroke population.

## 2. Materials and Methods

### 2.1. Study Design and Participants

Older adult patients with dysphagia and acute illnesses were randomly assigned to three parallel groups: the thickeners group, the swallowing exercises group, and the control group. The study was conducted at a comprehensive medical ward in a medical center in Taiwan, and the intervention period for the thickeners and swallowing exercises groups lasted three weeks. Demographic characteristics, the Charlson Comorbidity Index (CCI) [20], acute illness at admission, income, activities of daily living (ADL), Spiritual Assessment Scale (SAS) [21], Chinese Happiness Inventory (CHI) [22], Physical Activity Scale for the Elderly (PASE) [23], and Eating Assessment Tool-10 (EAT-10) [24] score were assessed as variables. These assessments were conducted both after admission and before discharge.

The study employed a cluster randomized controlled trial sampling method. Patient recruitment followed these steps: (1) Screening Period: All ward admissions from October 2019 to August 2020 were screened, focusing on those aged 65 and older. (2) Swallowing Difficulty Assessment: Eligible patients were evaluated using the EAT-10 scoring system to gauge the severity of swallowing difficulties. (3) Eligibility Criteria: Patients with an EAT-10 score of 3 or higher were eligible, indicating the presence of swallowing difficulties. (4) Exclusion Criteria: Eligible patients were further screened for exclusion criteria, including end-stage illness, inability to comply with interventions, or refusal to participate. This systematic process ensured the selection of an appropriate patient population for the cluster randomized controlled trial, considering age and the presence of swallowing difficulties, while excluding those not meeting the study’s criteria or unwilling to participate. A flowchart depicting the patient selection process is shown in Figure 1. Written informed consent was obtained from all participants, and the study was approved by the Institutional Review Board (IRB) of Kaohsiung Chang Gung Memorial Hospital (IRB number: 202200244B0C502).

### 2.2. Randomized

The study utilized a cluster randomization design, wherein allocation was kept concealed from ward rooms. The study nurse conducted visits to these ward rooms while remaining unaware of the allocation until the baseline assessment was completed. Blinding was not feasible due to the nature of the intervention. All ward rooms were randomized simultaneously using sealed envelopes. Cluster randomization of ward rooms was conducted, employing a block length of three and stratified by region, resulting in a 1:1:1 assignment to either the intervention or control group. The randomization process was carried out by a member of the study team who was blinded to patient assignment and the sampling process. Data collection and subsequent analysis were conducted in a blinded manner regarding the group allocation of the patients to minimize bias and ensure the integrity of the study’s results.

### 2.3. Interventions

#### 2.3.1. Control Group

Patients in the control group received standard care, which was defined as personalized dietary counseling provided by a nutritionist. The nutritionist assessed each patient’s needs and made recommendations to improve and maintain an adequate calorie intake.

#### 2.3.2. Intervention Groups

The swallowing exercises group received a combination of individual dietary counseling (standard care) and personalized interventions, including stretching exercises to improve tongue, chin, and throat mobility, as well as compensatory and swallowing techniques to enhance food intake normalization. The patients underwent training twice daily for a minimum of 10 min each session. Speech therapists educated the patients and their families, and the clinical nurse provided guidance during one-hour sessions twice daily for three weeks. The exercises aimed to optimize swallowing function and safety.

In the thickeners group, personalized dietary counseling (standard care) was complemented with the use of commercial thickening powder formulations, including xanthan gum, maltodextrin, and potassium chloride [25], to prepare thickened liquids. The interventions in this group were as follows. (1) Adjusting the texture of liquids using thickening agents to aid swallowing. (2) Modifying the consistency of food by grinding or cutting it into smaller pieces to enhance swallowing safety. (3) Providing oral hygiene recommendations to ensure oral health and reduce the risk of infection. (4) Employing positional changes, such as rotating the chin and head to the affected side, to promote more effective swallowing.

### 2.4. Measures

A socio-demographic schedule was used to collect information on gender, age, marital status, educational level, religion, household income, frailty score, body mass index, and acute illness. The EAT-10, a self-administered screening tool, was used to assess swallowing difficulties. The EAT-10 comprises 10 items, with each item scored from 0 to 4. The total score ranges from 0 to 40, with higher scores indicating more severe swallowing issues. A score of 3 or higher on the EAT-10 is indicative of swallowing impairment [26]. The SAS used in this study was based on the translation by Zhao et al. of the Spiritual Needs Questionnaire (SpNQ) developed by Professor Büssing [27,28]. This tool focuses on the non-religious aspects of spirituality, centering on existence, relationships, and awareness. It comprises 27 items that categorize spirituality into six dimensions: Inner Peace Needs, Giving Needs, Belief-Blessing Needs, Belief-Resource Needs, Existential-Reflective Needs, and Existential-Acceptance Needs. Participants rate all items on a 4-point Likert scale, with higher scores indicating greater levels of spiritual need in each respective dimension. The Chinese version of the complete SpNQ exhibited a Cronbach’s α coefficient of 0.910, and the Cronbach’s α coefficients for each dimension ranged from 0.661 to 0.965.

In this study, the CHI developed by Lu et al. was used to evaluate an individual’s subjective perception of happiness. The CHI consists of 20 items and encompasses three dimensions: positive affect, negative affect, and life satisfaction. It covers various aspects such as work achievements, social comparison, inner peace, optimism, social commitment, positive emotions, sense of control, physical health, and self-satisfaction. Participants rate the items on a 4-point Likert scale ranging from 0 “disagree” to 3 “strongly agree.” The total CHI score ranges from 0 to 60, with higher scores indicating a higher overall level of happiness [29]. The validated Chinese version of the PASE questionnaire was used to assess physical activity levels [30]. The PASE is a widely used measure in epidemiological research to assess physical activity in older adults, and it has been proven to be reliable and valid in the Chinese population [31]. The tool consists of 12 self-reported items that cover leisure, household, and work-related activities during the previous week. Leisure-time activities are rated on two ordinal scales: (1) the number of days engaged in the activity (0 = never to 3 = often), and (2) the number of hours per day spent on the activity (<1 to >4 h) to determine the frequency value (hours per day) for each activity. Household and work-related activities are assessed as “yes” or “no.” A score of 1 is given for “yes” responses, and 0 for “no” responses. The scores for the 12 activities are individually weighted based on their corresponding intensity and then summed to obtain the total PASE score, which ranges from 0 to 400 or higher. A higher score indicates a greater level of physical activity [30].

### 2.5. Sampling

G Power analysis was performed to calculate the minimum sample size, taking into account a repeated measure analysis of variance (ANOVA) with an F-test (ANOVA repeated measures, within-between interaction). There were three groups in the study, with two repeated measurements. The correlation between repeated measurements was set at 0.5. A conservative estimate of a low effect size of 0.25 was used, with a significance level (α) of 0.05 and a statistical power of 0.8. Sample size calculation was performed using a conservative estimation approach, resulting in an estimated required sample size of 42 participants. Considering an anticipated dropout rate of 20%, the target recruitment was set at 50 participants. The cluster randomization of ward rooms was carried out with a block length of three and stratification by region, leading to an equitable 1:1:1 assignment of patients to either the intervention or control group within each cluster. Consequently, the sample size within each group is deemed to be sufficient for the study’s purposes.

### 2.6. Statistical Analysis

To compare the three groups (thickeners group, swallowing exercises group, and control group), all statistical analyses were performed using SPSS 22.0 (SPSS Inc., Armonk, NY, USA). Cross-tabulation and chi-square tests were used to compare categorical variables and assess intergroup differences in baseline data. The baseline data were found to be homogeneous across the groups. To evaluate the effectiveness of the intervention measures between the thickeners and swallowing exercises groups and the control group, we employed generalized estimating equations for inferential analysis of repeated measurements. A *p* value of less than 0.05 was statistically significant.

## 3. Results

Of 236 initially screened patients, 124 met the inclusion criteria and were included in the study. After applying the exclusion criteria, the final study population consisted of 72 patients, with 24 in each group (control group, thickeners group, and swallowing exercises group) (Figure 1). The actual sample size (*n* = 72) exceeded the required size.

The mean ages of the swallowing exercises group, thickeners group, and control group were 72.46, 72.13, and 70.88 years, respectively. There were no significant differences among the groups in terms of demographic characteristics, including age, gender, marital status, education level, religious beliefs, family income, frailty, body mass index, CCI, and reasons for hospitalization (Table 1). Prior to the intervention, there were no significant differences among the groups in mental well-being, happiness, physical activity level, eating assessment, ADL, and daily caloric intake (Table 2). However, after the intervention, significant differences were observed among the groups in spirituality (*f*: 10.680, *p* < 0.001), happiness (*f*: 18.046, *p* < 0.001), physical activity level (*f*: 5.296, *p* = 0.007), EAT (*f*: 3.268, *p* = 0.044), ADL (*f*: 3.997, *p* = 0.023), and daily caloric intake (*f*: 3.899, *p* = 0.025).

There were no significant differences in EAT, PASE, happiness, and spirituality scores among the control group, thickeners group, and swallowing exercises group at baseline (Table 3; Appendix A). However, following the intervention, both the thickeners and swallowing exercises groups exhibited significant improvements in these outcome measures compared to the control group. The effectiveness of the interventions and the overall time had a positive effect on the performance, which was further supported by generalized estimating equation analysis. Specifically, the thickeners and swallowing exercises groups showed significant improvements in EAT scores (Wald *χ*^2^: 17.56; *p* < 0.001), indicating improved swallowing function, compared to the control group. Similarly, the intervention groups showed significant improvements in PASE scores (Wald *χ*^2^: 9.09; *p* = 0.011), indicating increased physical activity levels, compared to the control group. Moreover, the thickeners and swallowing exercises groups showed notable improvements in happiness (Wald *χ*^2^: 19.31; *p* < 0.001) and spirituality scores (Wald *χ*^2^: 8.87; *p* = 0.012) compared to the control group, suggesting positive effects on emotional well-being and spiritual experiences.

In our further analysis, we examined the interaction between group and time, using the control group at post-intervention as the reference point. The findings revealed significant improvements in EAT scores for both the swallowing exercises group (β: −1.38, Wald *χ*^2^: 11.39, *p* = 0.001) and the thickeners group (β: −1.92, Wald *χ*^2^: 13.94, *p* < 0.001). Furthermore, there were notable enhancements in PASE scores after the intervention for both intervention groups (β: 52.44, Wald *χ*^2^: 9.08, *p =* 0.003; β: 37.75, Wald *χ*^2^: 4.03, *p* = 0.045), as well as in Happiness scores after the intervention (β: 11.44, Wald *χ*^2^: 10.91, *p* = 0.001; β: 14.44, Wald *χ*^2^: 15.48, *p* < 0.001). Finally, there were significant improvements in Spirituality scores after the intervention (β: 20.42, Wald *χ*^2^: 5.90, *p* = 0.015; β: 24.06, Wald *χ*^2^: 7.09, *p* = 0.008) for both intervention groups. However, there were no significant differences between the thickeners group and the swallowing exercises group in terms of the improvements observed after the intervention.

## 4. Discussion

This randomized controlled trial was designed to explore the differences among three parallel groups: the thickeners group, the swallowing exercises group, and the control group, concerning spiritual well-being, physical activity, and happiness in older adult patients experiencing swallowing difficulties during acute hospitalization. Based on the results of our statistical analyses, it was evident that both interventions, as opposed to the control group, had a positive impact on enhancing swallowing function and calorie intake. Furthermore, these interventions were also associated with improvements in spiritual well-being, physical activity, and happiness among older adult patients during their acute hospitalization. This study provides evidence supporting the effectiveness of early intervention with thickeners or swallowing exercises in improving swallowing function and calorie intake in older adult patients during acute hospitalization. These findings are consistent with the results of a study conducted by Rofes et al., which also demonstrated that thickeners could reduce oral residue and improve swallowing difficulties [32]. In addition, our results indicated that performing swallowing exercises for one hour twice daily over three weeks could also improve swallowing function as well as calorie intake, consistent with other studies involving different patient populations [33].

Prior studies have not investigated the association between interventions such as thickeners or swallowing exercises and measures of happiness and spirituality. However, our results showed significant improvements in these aspects, and both the thickeners and swallowing exercises groups exhibited better swallowing function compared to the control group, along with enhanced functional capacity. Furthermore, both intervention groups showed notable increases in happiness and spirituality scores compared to the control group. These results indicate that interventions involving thickeners and swallowing exercises may improve both swallowing function and functional capacity and positively impact happiness and spirituality.

Both the thickeners and swallowing exercises groups demonstrated significant improvements in daily calorie intake, activity level, happiness, and spirituality compared to the control group after one month of the intervention, which is in line with prior research [34]. Previous research has examined the efficacy of thickeners and swallowing exercises in improving swallowing difficulties in stroke patients. Yan et al. reported that the use of thickeners improved swallowing difficulties and maintained calorie intake after 7 days of hospitalization [35]. In addition, Wang et al. reported that a 4-week regimen of swallowing exercises increased muscle strength and ameliorated swallowing difficulties [36], and Farpour et al. reported diverse levels of improvement over time with various swallowing interventions [37]. However, it is still unclear which of these interventions is most beneficial. A recent meta-analysis did not find a conclusive difference in effectiveness between swallowing training and thickeners in improving swallowing difficulties in stroke patients [8], which is consistent with our findings. Short-term studies have demonstrated comparable outcomes between swallowing exercises and the use of thickeners in enhancing swallowing function, calorie intake, daily activities, and mobility. Taken together, these results suggest that improvements in tongue muscle strength through swallowing exercises may necessitate a longer duration to manifest. To ascertain potential disparities in long-term effectiveness, further research with extended follow-up periods is required.

## 5. Conclusions

Interventions involving thickeners or swallowing exercises enhanced calorie intake and mobility among older adult patients during acute hospitalization, which in turn contributed to improvements in spirituality and happiness. Nevertheless, additional research is needed to comprehensively understand the long-term effects of these interventions.

## 6. Limitations and Recommendations for Future Studies

Our study has several limitations. First, it was conducted at a single medical center with a relatively small sample size, which may restrict the generalizability of the findings to other regions or populations. Additionally, the use of the self-assessment tool EAT-10 instead of video-fluoroscopic swallowing assessments may have introduced subjective bias in the results. Moreover, the study primarily focused on short-term intervention effects during acute hospitalization in older adult patients, making it challenging to draw definitive conclusions about long-term outcomes. To address these limitations and gain a comprehensive understanding, further research should explore different intervention methods and assess their long-term effectiveness in diverse settings and patient groups.

## Figures and Tables

**Figure 1 healthcare-11-02595-f001:**
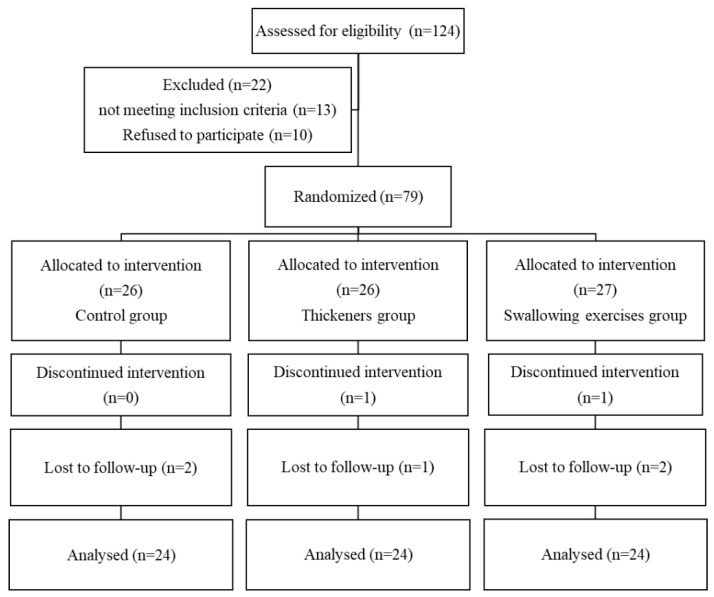
Consort flow diagram of the participants in this study.

**Table 1 healthcare-11-02595-t001:** Characteristics of the enrolled patients before the interventions.

Variable	Control Group	Thickeners Group	Swallowing Exercises Group	*t/f*	*p*
N	24	24	24		
Gender (%)				1.493	0.288
Male	12 (50.0)	16 (79.2)	17 (70.8)		
Female	12 (50.0)	8 (20.8)	7 (29.2)		
Age (years), median (interquartile)	70.88 (66.0, 74.0)	72.13 (66.0, 75.8)	72.46 (66.0, 78.8)	0.417	0.661
Marital status (%)				0.001	0.854
Single	2 (8.3)	3 (12.5)	2 (8.3)		
With spouse	22 (91.7)	21 (87.5)	22 (91.7)		
Educational level (%)				0.750	0.421
Below elementary school	13 (54.2)	9 (37.5)	11 (45.8)		
Junior and senior high school	10 (41.7)	15 (62.5)	11 (45.8)		
Above college	1 (4.1)	0 (0)	2 (8.4)		
Religion (%)				1.611	0.248
Yes	22 (91.7)	21 (87.5)	18 (75.0)		
No	2 (8.3)	3 (12.5)	6 (25.0)		
Household income USD (%)				0.383	0.562
Below 1000	7 (29.2)	4 (16.6)	7 (29.2)		
1000~1700	12 (50.0)	10 (41.7)	10 (41.7)		
More than 1700	5 (20.8)	10 (41.7)	7 (29.2)		
Frailty score (%)				1.105	0.545
Mildly	9 (37.6)	7 (29.2)	8 (33.3)		
Moderately	11 (45.8)	10 (41.6)	7 (29.2)		
Severely	4 (16.6)	7 (29.2)	9 (37.5)		
BMI (kg/m^2^) Median (interquartile)	25.13 (22.3, 28.5)	23.30 (21.7, 26.9)	22.51 (20.7, 25.6)	2.712	0.074
<18.5 (%)	1 (4.2)	3 (12.5)	2 (8.3)		
CCI Median (interquartile)	4.96 (3.3, 7.0)	5.88 (4.0, 7.8)	5.21 (4.0, 5.0)	1.941	0.151
>5 (%)	13 (54.2)	15 (62.5)	17 (70.8)		
Acute illness (%)				1.487	0.142
Infectious diseases	7 (29.2)	7(29.2)	13 (54.2)		
Gastrointestinal diseases	7 (29.2)	2 (8.3)	4 (16.7)		
Cardiovascular diseases	10 (41.7)	15 (62.5)	7 (29.2)		

BMI, Body Mass Index; CCI, Charlson Comorbidity Index.

**Table 2 healthcare-11-02595-t002:** Scales and daily calorie intake before and after the interventions. Improving spirituality, happiness, physical activity level, swallowing function, activity of daily living, and daily caloric intake after interventions in thickeners and swallowing exercises groups.

Variable	Control Group	Thickeners Group	Swallowing Exercises Group	*f*	*p*
T0					
Spirituality, median (interquartile)	78.79 (72.8, 88.8)	90.25 (76.5, 103.8)	86.33 (67.5, 103.3)	2.707	0.074
Happiness, median (interquartile)	15.63 (6.5, 22.0)	16.41 (5.0, 26.5)	17.95 (6.0, 30.3)	0.232	0.794
PASE, median (interquartile)	68.38 (5.3, 132.8)	84.44 (8.6, 156.8)	71.50 (0, 127.1)	0.359	0.700
High (%)	6 (25.0)	7 (29.2)	6 (25.0)		
Moderate (%)	4 (16.7)	4 (16.7)	5 (20.8)		
Low (%)	14 (58.3)	13 (54.2)	13 (54.2)		
EAT, median (interquartile)	6.33 (4.0, 9.0)	6.63 (3.0, 9.8)	6.25 (3.0, 9.0)	0.109	0.897
ADL, median (interquartile)	62.29 (45.0, 85.0)	59.38 (41.3, 73.8)	62.08 (45.0, 90.0)	0.135	0.874
Daily calories, median (interquartile)	1346 (1120, 1569)	1351 (1205, 1450)	1300 (1015, 1527)	0.211	0.811
T1					
Spirituality, median (interquartile)	100.17 (86.3, 119.5)	135.68 (118.3, 164.8)	128.12 (114.3, 144.8)	10.680	<0.001
Happiness, median (interquartile)	34.95 (27.0, 41.0)	50.19 (45.5, 57.5)	48.73 (43.0, 54.0)	18.046	<0.001
PASE, median (interquartile)	68.56 (7.5, 117.0)	122.38 (61.5, 174.0)	124.13 (60.4, 214.5)	5.296	0.007
High (%)	3 (12.5)	14 (58.3)	9 (37.5)		
Moderate (%)	11 (45.8)	3 (12.5)	7 (29.2)		
Low (%)	10 (41.7)	7 (29.2)	8 (33.3)		
EAT, median (interquartile)	6.33 (4.0, 9.0)	4.71 (3.0, 7.5)	4.88 (3.0, 6.8)	3.268	0.044
ADL, median (interquartile)	67.71 (50.0, 90.0)	75.63 (61.5, 174.0)	80.83 (67.5, 93.8)	3.997	0.023
Daily calories, median (interquartile)	1462 (1290, 1650)	1664 (1582, 1835)	1637 (1265, 1772)	3.899	0.025

T0, Before intervention; T1, After intervention; PASE, Physical Activity Scale for the Elderly; EAT, Eating Assessment Tool; ADL, Activity of daily living.

**Table 3 healthcare-11-02595-t003:** Generalized estimating equation analysis of the enrolled patients. Both the thickeners and swallowing exercises groups exhibited significant improvements in spirituality, happiness, physical activity level, swallowing function, activity of daily living, compared to the control group.

Outcome Variables	*β*	*SE*	Wald *χ^2^*	*p*	95% CI
EAT					
Intercept (CG T0)	6.33	0.53	141.45	<0.001	5.29, 7.38
Group (SG vs. CG in T0)	−0.08	0.81	0.01	0.918	−1.67, 1.50
Group (TG vs. CG in T0)	0.29	0.81	0.13	0.718	−1.29, 1.88
Time overall (CG)			32.26	<0.001	
Group × time overall			17.56	<0.001	
Group × time (SG vs. T1)	−1.38	0.41	11.39	0.001	−2.17, −0.58
Group × time (TG vs. T1)	−1.92	0.51	13.94	<0.001	−2.92, −0.91
PASE					
Intercept (CG T0)	68.38	13.34	26.26	<0.001	42.22, 94.53
Group (SG vs. CG in T0)	3.13	19.53	0.03	0.873	−35.16, 41.41
Group (TG vs. CG in T0)	16.06	19.44	0.68	0.409	−22.04, 54.16
Time overall (CG)			18.32	<0.001	
Group × time overall			9.09	0.011	
Group × time (SG vs. T1)	52.44	17.41	9.08	0.003	18.32, 86.55
Group × time (TG vs. T1)	37.75	18.80	4.03	0.045	0.91, 74.59
Happiness					
Intercept (CG T0)	15.63	2.04	58.75	<0.001	11.63, 19.62
Group (SG vs. CG in T0)	2.33	3.28	0.06	0.144	−4.10, 8.76
Group (TG vs. CG in T0)	0.80	3.30	0.06	0.810	−5.66, 7.25
Time overall (CG)			340.55	<0.001	
Group × time overall			19.31	<0.001	
Group × time (SG vs. T1)	11.44	3.46	10.91	0.001	4.65, 18.23
Group × time (TG vs. T1)	14.44	3.67	15.48	<0.001	7.25, 21.64
Spirituality					
Intercept (CG T0)	78.79	3.01	682.69	<0.001	72.88, 84.70
Group (SG vs. CG in T0)	7.54	5.08	2.21	0.138	−2.41, 17.50
Group (TG vs. CG in T0)	11.46	4.40	6.79	0.009	2.84, 20.08
Time overall (CG)			101.08	<0.001	
Group × time overall			8.87	0.012	
Group × time (SG vs. T1)	20.42	8.41	5.90	0.015	3.94, 36.90
Group × time (TG vs. T1)	24.06	9.04	7.09	0.008	6.34, 41.77

T0, Before intervention; T1, After intervention; EAT, Eating Assessment Tool; PASE, Physical Activity Scale for the Elderly; CG, Control group; TG, Thickeners group; SG, Swallowing exercises group.

## Data Availability

The data associated with the paper are not publicly available but are available from the corresponding author on reasonable request.

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
