# Peer review of "Enhancing Spiritual Well-Being, Physical Activity, and Happiness in Hospitalized Older Adult Patients with Swallowing Difficulties: A Comparative Study of Thickeners and Swallowing Exercises"

_healthcare, 2023, doi:10.3390/healthcare11182595_

Round 1
Reviewer 1 Report
- line 74: What hypothesis could be formulated for this study considering the population investigated?
- line 82 a 84: The Charlson Comorbidity Index, Spiritual Assessment Scale (SAS), Chinese Happiness Inventory (CHI), Physical Activity Scale for the Elderly (PASE), and Eating Assessment Tool-10 (EAT-10) score. I suggest that all these instruments used for the analyses of this study be referenced, preferably with studies with a similar population or investigating the same approach/effect.
- line 162-169: A sample size calculation to size the sample size required for this study is important to be performed and informed in the Statistical Analysis session.
- line 191 and 209: Enter the significance value of p in the captions of Tables 1, 2 and 3.
-
Author Response
Re: Enhancing Spiritual Well-being, Physical Activity, and Happiness in Hospitalized Old Adult Patients with Swallowing Difficulties: A Comparative Study of Thickeners and Swallowing Exercises
Response to reviewers’ comments
We thank the reviewers for their helpful comments and for giving us the chance to revise our manuscript. The changes are marked in blue.
#Reviewer: 1
Comment 1: line 74: What hypothesis could be formulated for this study considering the population investigated?
Response: Thank you for your suggestion. We add a new phrase to address the aim of our study to investigate the spiritual well-being, physical activity, and happiness impact of swallowing interventions in older adult patients with dysphagia beyond the stroke population. Please see P2-3. Lines 76-98.
Comment 2: line 82-84: The Charlson Comorbidity Index, Spiritual Assessment Scale (SAS), Chinese Happiness Inventory (CHI), Physical Activity Scale for the Elderly (PASE), and Eating Assessment Tool-10 (EAT-10) score. I suggest that all these instruments used for the analyses of this study be referenced, preferably with studies with a similar population or investigating the same approach/effect.
Response: We agree to adopt your suggestions. All these instruments used with studies were referenced. Please see P3. Lines 106-108.
Comment 3: line 162-169: A sample size calculation to size the sample size required for this study is important to be performed and informed in the Statistical Analysis session.
Response: Thank you for your suggestion. We add detailed description as “Sample size estimation for this study was performed using G Power 3.1.9 software, taking to account a repeated measure analysis of variance (ANOVA) with an F-test (ANOVA repeated measures, within-between interaction). There were three groups in the study, with two repeated measurements. The correlation between repeated measurements was set at 0.5. A conservative estimate of a low effect size of 0.25 was used, with a significance level (α) of 0.05 and a statistical power of 0.8. Sample size calculation was performed using a conservative estimation approach, resulting in an estimated required sample size of 42 participants. Considering an anticipated dropout rate of 20%, the target recruitment was set at 50 participants. The cluster randomization of ward rooms was carried out with a block length of three and stratification by region, leading to an equitable 1:1:1 assignment of patients to either the intervention or control group within each cluster. Consequently, the sample size within each group is deemed to be sufficient for the study's purposes.” in the Statistical Analysis session. Please see P5. Lines 207-219.
Comment 4: line 191 and 209: Enter the significance value of p in the captions of Tables 1, 2 and 3.
Response: We agree your suggestion. The significance of findings has been put in the captions of Tables 2 and 3. Please see P7, Lines 256-258 and P9, Line 316-318.

Reviewer 2 Report
Thank you for submitting the manuscript, entitled “Enhancing Spiritual Well-being, Physical Activity, and Happiness in Hospitalized Elderly Patients with Swallowing Difficulties: A Comparative Study of Thickeners and Swallowing Exercises” to Healthcare. Please see my comments below.
Abstract:
· Was this study randomized controlled study?
· What was the sampling method?
Introduction
· The authors should elaborate more about spiritual well-being, physical activity, and happiness elderly patients. By the way, we won’t use “elderly”.
· The flow of idea on study areas is not clear. The knowledge gaps focus on Spiritual Well-being, Physical Activity, and Happiness in Hospitalized Elderly Patients with Swallowing Difficulties should be elaborated more.
Material and methods
· What was the study design?
· How to conduct the randomization? Who did it?
· What about the sampling method?
· Any sample size calculation for each group?
· How to prevent the contamination of the interventions?
· There were 3 interventions. Therefore, the heading should be “Interventions”
· Who implemented the interventions? Two interventions were implemented by the same staff?
· The authors should do more analysis to identify the effects of the intervention compared to the control group. The results should include the differences of the effects between 2 interventions.
Discussion
· The sample size calculation should be moved to Method
· The sample size calculation should be for each group. How did you calculate the sample size?
· The interpretation of the result is inadequate. It’s not enough to only report the significance of the intervention.
Minor revision on English writing.
Author Response
Re: Enhancing Spiritual Well-being, Physical Activity, and Happiness in Hospitalized Old Adult Patients with Swallowing Difficulties: A Comparative Study of Thickeners and Swallowing Exercises
Dear reviewer,
Thank you for your good comments and giving us chance to improve our manuscript. The revisions parts are in blue print in the revised manuscript.
#Reviewer 2:
Abstract:
Comment 1: Was this study randomized controlled study?
Response: This study was randomized controlled study. Please see P1.
Comment 2: What was the sampling method?
Response: The sampling method was a random cluster sampling approach was used, with each three ward rooms assigned to the thickeners group, swallowing exercises group and control group. Please see P1.
Introduction
Comment 3: The authors should elaborate more about spiritual well-being, physical activity, and happiness elderly patients. By the way, we won’t use “elderly”.
Response: We agree to adopt your professional suggestion and change “elderly patients” to “older adult patients” in the revised edition.
Comment 4: The flow of idea on study areas is not clear. The knowledge gaps focus on Spiritual Well-being, Physical Activity, and Happiness in Hospitalized Elderly Patients with Swallowing Difficulties should be elaborated more.
Response: Thank you for your suggestion. We add a new phrase as “Interventional strategies for dysphagia are frequently employed in patients with Parkinson's disease, stroke, multiple sclerosis, and oral cancer [8,14]. However, most previous research studies have focused on evaluating the effectiveness of these strategies specifically in stroke patients, with relatively few focusing on hospitalized older adult patients with swallowing difficulties. Consequently, the effectiveness of implementing these interventions early in hospitalized older adult patients with dysphagia in terms of reducing adverse outcomes remains uncertain. To the best of our knowledge, there were no controlled studies exploring the effect of swallowing interventions on spiritual well-being, physical activity, and happiness of older adult patients with swallowing difficulties during acute hospitalization.
Research has indicated that older adult patients during acute hospitalization are at an increased risk of malnutrition by 155% when they experience swallowing difficulties, while the risk of diminished activities of daily living rises by 60% [15]. It is evident that swallowing difficulties have a substantial impact on nutritional intake and daily functioning in these population. Following a decline in activities of daily living, older adults often struggle to maintain social engagement, significantly affecting their spiritual and psychological well-being. Figueira et al. have demonstrated numerous benefits of regular physical activity interventions for the older adults, not only enhancing muscle mass but also improving spiritual well-being, potentially promoting healthier aging [16]. Moreover, increased physical activity has been found to enhance social engagement and lead to more pronounced improvements in spiritual well-being [17]. Several studies have also suggested that enhanced physical activity in the elderly indirectly contributes to an improved sense of happiness [18,19]. However, it remains uncertain whether improvements in swallowing function following the onset of swallowing difficulties in older adult patients during acute hospitalization would result in a subsequent enhancement of their activity levels, spiritual well-being, and happiness. To address this knowledge gap, the aim of this study was to investigate the spiritual well-being, physical activity, and happiness impact of these interventions in older adult patients with dysphagia beyond the stroke population.” Please see P2-3. Lines 70-98.
Material and methods
Comment 5: What was the study design?
Response: The study employed a cluster randomized controlled trial sampling method. Patient recruitment followed these steps: 1) Screening Period: All ward admissions from October 2019 to August 2020 were screened, focusing on those aged 65 and older. 2) Swallowing Difficulty Assessment: Eligible patients were evaluated using the EAT-10 scoring system to gauge the severity of swallowing difficulties. 3) Eligibility Criteria: Patients with an EAT-10 score of 3 or higher were eligible, indicating the presence of swallowing difficulties. 4) Exclusion Criteria: Eligible patients were further screened for exclusion criteria, including end-stage illness, inability to comply with interventions, or refusal to participate. This systematic process ensured the selection of an appropriate patient population for the cluster randomized controlled trial, considering age and the presence of swallowing difficulties, while excluding those not meeting the study's criteria or unwilling to participate. Please see P3. Lines 111-122.
Comment 6: How to conduct the randomization? Who did it?
Response: The study utilized a cluster randomization design, wherein allocation was kept concealed from ward rooms. The study nurse conducted visits to these ward rooms while remaining unaware of the allocation until the baseline assessment was completed. Blinding was not feasible due to the nature of the intervention. All ward rooms were randomized simultaneously using sealed envelopes. Cluster randomization of ward rooms was conducted, employing a block length of three and stratified by region, resulting in a 1:1:1 assignment to either the intervention or control group. The randomization process was carried out by a member of the study team who was blinded to patient assignment and the sampling process. Data collection and subsequent analysis were conducted in a blinded manner regarding the group allocation of the patients to minimize bias and ensure the integrity of the study's results. Please see P3. Lines 126-137.
Comment 7: What about the sampling method?
Response: The study employed a cluster randomized controlled trial sampling method. Patient recruitment followed these steps: 1) Screening Period: All ward admissions from October 2019 to August 2020 were screened, focusing on those aged 65 and older. 2) Swallowing Difficulty Assessment: Eligible patients were evaluated using the EAT-10 scoring system to gauge the severity of swallowing difficulties. 3) Eligibility Criteria: Patients with an EAT-10 score of 3 or higher were eligible, indicating the presence of swallowing difficulties. 4) Exclusion Criteria: Eligible patients were further screened for exclusion criteria, including end-stage illness, inability to comply with interventions, or refusal to participate. This systematic process ensured the selection of an appropriate patient population for the cluster randomized controlled trial, considering age and the presence of swallowing difficulties, while excluding those not meeting the study's criteria or unwilling to participate. Please see P3. Lines 111-122.
Comment 8: Any sample size calculation for each group?
Response: In this study, we did not calculate the sample size individually for each group. Instead, we employed G Power to determine the overall sample size. The cluster randomization of ward rooms was carried out with a block length of three and stratification by region, leading to an equitable 1:1:1 assignment of patients to either the intervention or control group within each cluster. Consequently, the sample size within each group is deemed to be sufficient for the study's purposes. Please see P5. Lines 207-219.
Comment 9: How to prevent the contamination of the interventions?
Response: To prevent intervention contamination, we implemented a random cluster sampling method. In this approach, every three ward rooms were assigned to one of three groups: the thickeners group, the swallowing exercises group, and the control group. This allocation ensured a clear separation between the intervention and control groups, reducing the risk of cross-contamination between the groups. Please see P3. Lines 126-137.
Comment 10: There were 3 interventions. Therefore, the heading should be “Interventions”
Response: Thank for your suggestion. We have rewritten the heading. Please see P3. Lines 138.
Comment 11: Who implemented the interventions? Two interventions were implemented by the same staff?
Response: Patients in the control group received standard care, which was characterized by personalized dietary counseling administered by a nutritionist. The nutritionist conducted individual assessments to determine each patient's dietary requirements and offered recommendations aimed at enhancing and sustaining an appropriate calorie intake. In contrast, for patients in the thickeners group, the intervention was overseen by clinical nurses. These clinical nurses, during two one-hour sessions daily for a duration of three weeks, provided guidance and implemented the use of thickeners to assist patients with swallowing difficulties. Additionally, speech therapists played a vital role by educating both patients and their families, ensuring a comprehensive approach to managing swallowing difficulties within the study. Please see P3-4. Lines 139-161.
Comment 12: The authors should do more analysis to identify the effects of the intervention compared to the control group. The results should include the differences of the effects between 2 interventions.
Response: Thanks for your suggestion. Although two different intervention measures (thickeners and swallowing exercises) were employed, various statistical tests including cross-tabulation, chi-square tests, and generalized estimating equations were used. These analyses revealed that both intervention measures were effective; however, no significant difference was observed between the two interventions.
To be clarified, we add a new phrase as “In our further analysis, we examined the interaction between group and time, using the control group post-intervention as the reference point. The findings revealed significant improvements in EAT scores for both the swallowing exercises group (β: -1.38, Wald χ2: 11.39, p= .001) and the thickeners group (β: -1.92, Wald χ2: 13.94, p< .001). Furthermore, there were notable enhancements in PASE scores after the intervention for both intervention groups (β: 52.44, Wald χ2: 9.08, p= .003; β: 37.75, Wald χ2: 4.03, p= .045), as well as in Happiness scores after the intervention (β: 11.44, Wald χ2: 10.91, p= .001; β: 14.44, Wald χ2: 15.48, p< .001). Finally, there were significant improvements in Spirituality scores after the intervention (β: 20.42, Wald χ2: 5.90, p: .015; β: 24.06, Wald χ2: 7.09, p= .008) for both intervention groups.” Please see P8-9. Lines 304-313.
Discussion
Comment 13: The sample size calculation should be moved to Method.
Response: Thank for your suggestion. We have moved the sample size calculation to the Statistical Analysis session. Please see P5. Lines 207-219.
Comment 14: The sample size calculation should be for each group. How did you calculate the sample size?
Response: The G Power analysis was performed to calculate the minimum sample size. Taking to account a repeated measure analysis of variance (ANOVA) with an F-test (ANOVA repeated measures, within-between interaction). There were three groups in the study, with two repeated measurements. The correlation between repeated measurements was set at 0.5, and a nonsphericity correction value of 1 was assumed. A conservative estimate of a low effect size of 0.25 was used, with a significance level (α) of 0.05 and a statistical power of 0.8. Sample size calculation was performed using a conservative estimation approach, resulting in an estimated required sample size of 42 participants. Considering an anticipated dropout rate of 20%, the target recruitment was set at 50 participants. The cluster randomization of ward rooms was carried out with a block length of three and stratification by region, leading to an equitable 1:1:1 assignment of patients to either the intervention or control group within each cluster. Consequently, the sample size within each group is deemed to be sufficient for the study's purposes. Please see P5. Lines 207-219.
Comment 15: The interpretation of the result is inadequate. It’s not enough to only report the significance of the intervention.
Response: Thank for your suggestion. This randomized controlled trial was designed to explore the differences among three parallel groups: the thickeners group, the swallowing exercises group, and the control group, concerning spiritual well-being, physical activity, and happiness in older adult patients experiencing swallowing difficulties during acute hospitalization. Based on the results of our statistical analyses, it was evident that both interventions, as opposed to the control group, had a positive impact on enhancing swallowing function and calorie intake. Furthermore, these interventions were also associated with improvements in spiritual well-being, physical activity, and happiness among older adult patients during their acute hospitalization. Please see P9. Lines 323-331. We also mention that the study primarily focused on short-term intervention effects during acute hospitalization in older adult patients, making it challenging to draw definitive conclusions about long-term outcomes. To address these limitations and gain a comprehensive understanding, further research should explore different intervention methods and assess their long-term effectiveness in diverse settings and patient groups. Please see P10. Lines 379-383.
Comment 16: Comments on the Quality of English Language
Minor revision on English writing.
Response: Thank you for your suggestion. We sent the manuscript for English editing.

Round 2
Reviewer 2 Report
Good to see that the manuscript has been well revised. I have no further comment. very good.
The sample size calculation should be inserted in sampling section instead.
Author Response
#Reviewer: 2
Comment 1: Good to see that the manuscript has been well revised. I have no further comment. very good.
The sample size calculation should be inserted in sampling section instead.
Response: Thank you for your suggestion. The sample size calculation has been put in sampling section. Please see P5, Lines 199-212.
